# Mid–Long-Term Outcomes of Surgical Treatment of Legg-Calvè-Perthes Disease: A Systematic Review

**DOI:** 10.3390/children9081121

**Published:** 2022-07-27

**Authors:** Alessia Caldaci, Gianluca Testa, Eleonora Dell’Agli, Marco Sapienza, Andrea Vescio, Ludovico Lucenti, Vito Pavone

**Affiliations:** Department of General Surgery and Medical-Surgical Specialties, Section of Orthopaedics, A.O.U. Policlinico Rodolico-San Marco, University of Catania, 95123 Catania, Italy; alessia.c.92@hotmail.it (A.C.); gianpavel@hotmail.com (G.T.); dellagli.eleonora@hotmail.it (E.D.); marcosapienza09@yahoo.it (M.S.); andreavescio88@gmail.com (A.V.); ludovico.lucenti@gmail.com (L.L.)

**Keywords:** outcome, surgical treatment, Legg–Calvè–Perthes disease, pelvic osteotomy, femoral osteotomy, combined osteotomy

## Abstract

Background: Legg–Calvè–Perthes disease (LCPD) is a common childhood disease that usually occurs in 4- to 12-year-old children. Surgical treatment consists of femoral, pelvic, or combined osteotomies. This comprehensive review aimed to investigate the mid- and long-term outcome of the surgical treatment. Methods: A systematic review of PubMed, Science Direct, and MEDLINE databases was performed by two independent authors, using the keywords “outcome”, “surgical treatment”, “pelvic osteotomy”, “femoral osteotomy”, and “Legg–Calvè–Perthes disease” to evaluate studies of any level of evidence that reported the surgical outcome of LCPD. The result of every stage was reviewed and approved by two senior investigators. Results: A total of 2153 articles were found. At the end of the screening, we selected 23 articles eligible for full-text reading according to the inclusion and exclusion criteria. Our analysis showed that the main prognostic factors for surgical outcome in patients with LCPD are the age at onset and the degree of initial disease severity. Conclusions: Surgical treatment in patients older than 6 years has excellent results in Herring B and B/C hips and poor results in Herring C hips, with a slight advantage for patients between 6 and 8 years old.

## 1. Introduction

Legg-Calvè-Perthes disease (LCPD) is a common childhood disease resulting from avascular necrosis of the femoral head epiphysis in children with immature skeleton [1]. This condition usually occurs in children aged 4 to 12 years, and it is more frequent in males than females. Reported incidence rates in children <15 years vary, ranging from 0.2 per 100,000 to 19 per 100,000 [2].

The etiopathogenesis of LCPD was explained for the first time in 1910 by three physicians: Thornton Legg [3], Jacques Calvè [4], and George Perthes [5]. However, the etiology of LCPD remains unclear [6]. Skeletal dysplasias represent a very diverse group of diseases. The orthopedist plays a fundamental role in determining whether the disease is idiopathic or strongly correlated to a specific case. Complete documentation is the basis for correct clinical and radiological phenotypic characterization [7].

LCPD is an aseptic, non-inflammatory, and self-limiting process involving the interruption of the blood supply of the femoral head and acetabulum deformation. The first signs of the condition are pain, limpness, and limited mobility in the abduction and internal rotation of the hip joint [8]. It is assumed that the presence of alterations in the ossification centers and the presence of fragmentation of the epiphyseal growth plate are the basis of the development of hip pain in children with skeletal dysplasias [8,9,10,11,12,13,14,15,16,17,18].

Meanwhile, subluxation and then hip dislocation is observed in radiographic images. In adulthood, 30–50% of the patients experience symptoms involving the hip joint, similar to osteoarthritis [9]. The prognosis and treatment choice is made by age at onset, and Herring’s classification (lateral pillar classification) [9] clarifies the femoral head’s necrotic involvement. Stulberg’s classification [10] assesses the rating of residual femoral head deformity and joint congruence for the outcome.

Nowadays, both conservative and surgical treatments of LCPD raise questions; the first is reserved for patients <6 years, resorting to surgical treatment in severe cases of disease and late-onset. The common main goal is to preserve the femoral head, limiting the mechanical stress on the hip joint, which leads to its deformity [11]. The surgical treatments are varied and are focused on keeping the femoral head within the acetabulum. It consists of femoral, pelvic, or combined osteotomies.

Salter’s innominate osteotomy was the first pelvic redirection surgical technique used to treat LCPD in 1962 [12]. This technique redirects the acetabulum and provides anterolateral coverage of the femoral head. The common indications for Salter’s innominate osteotomy include >6 years old, Herring B, B/C, the initial fragmentation stage, suitable range of joint motion, and suitable joint congruence [12].

Nowadays, femoral varus osteotomy is the most common surgical treatment used worldwide. It was the first surgical procedure used to treat LCPD. This procedure aims to collocate the femoral head within the acetabulum and prevent osteoarthritis. The inclusion criteria to use this procedure are the same adopted for Salter’s innominate osteotomy. Possible consequences are different limb lengths and limping. For these reasons, it is recommended not to exceed 15° of varus [13,14].

However, in the most severe forms, other pelvic osteotomies (e.g., Chiary osteotomy, Triple innominate osteotomy, and lateral shelf acetabuloplasty) or combination treatments are indicated [15].

The aim of this review was to investigate the medium and long-term outcome of the surgical treatment and the prognostic factors that influence it.

## 2. Materials and Methods

### 2.1. Search Selection

A PIOS approach (Patient (P); Intervention (I); Outcome (O); and Study Design (S)) was utilized to carry out the study. The latest 23 studies assessing the surgical treatment (I) in LCPD-affected patients (P) were included. According to Stulberg, classification complications and total hip arthroplasties were then evaluated based on the accuracy (O). The following study designs (S) were included: randomized controlled trials (RCT), prospective (PS), retrospective (RS), case series (CS), case control (CC), and cohort (C) studies. The data currently available in the literature on the outcome of the surgical treatment of Perthes disease were analyzed. The article selection process was conducted in agreement with the Preferred Reporting Items for Systematic Review and Meta-Analyses (PRISMA) guidelines. It consists of a 27-item checklist and a flow chart divided into 4 steps. The checklist contains the essential items for transparent reporting of systematic reviews [16]. The study was conducted in January 2022 using PubMed, Science Direct, and MEDLINE databases. The search string included the terms outcome OR surgical treatment OR Legg–Calvè–Perthes disease. The analysis of the texts was carried out by two independent operators (A.C and E.D). Conflicts were resolved by consultation with two senior surgeons (VP and GT) (Table 1).

### 2.2. Study Selection

After searching the literature, 2153 articles were found. The analysis of the texts, conducted according to the inclusion and exclusion criteria, selected 23 eligible articles for the final analysis. A PRISMA flowchart of the selection and screening method is provided (Figure 1).

The inclusion criteria for the eligible articles were: (1) publication date from 2000 onwards, (2) Herring classification [40] for initial staging of severity, (3) Stulberg classification [10] for an outcome, (4) complete searchable article, and (5) follow-up of no less than 5 years.

The exclusion criteria were: (1) absence of initial assessment of severity according to Herring classification, (2) absence of Stulberg classification for the outcome, (3) follow-up of fewer than 5 years, and (4) unclear surgical technique used.

### 2.3. Data Extraction

Two reviewers (AC and ED) independently analyzed the articles. The extracted data were: number of affected hips, initial severity according to Herring classification [40] (Figure 2) (Herring A or B patients were classified as mild severity, B/C and C patients were classified as severe), age of onset, type of surgical treatment (femoral osteotomy, pelvic osteotomy, combined femoral and pelvic osteotomy and arthrodiastasis), average follow-up, radiographic outcome according to Stulberg’s classification [10], and the necessity of arthroplasty under the age of 50 years.

### 2.4. Outcome Measures

Patients with Stulberg classification I and II had a spherical congruency and were considered with suitable surgical outcomes, patients with Stulberg III were considered with fair surgical outcomes, and patients with Stulberg IV and V were considered with poor surgical outcomes.

## 3. Results

### 3.1. Demographics

Six hundred forty-nine hips were treated, with a mean age at onset of 9.3 years (range 6.5–10.9) and a mean follow-up of 9.6 years. Most (71%) were men, and 20% were women. The initial distribution according to Herring’s classification is described in Figure 2.

### 3.2. Surgical Treatment

Four articles looked at FVO [17,18,19,22], two articles evaluated the combination of FVO + Salter osteotomy [23,26], eight articles examined lateral shelf acetabuloplasty [20,24,27,28,29,33,36,37], two articles addressed arthrodiastasis [25,30], tow articles analyzed the Salter osteotomy [34,39], and two articles described the rotational transtrochanteric osteotomy [31,32]. Finally, one article analyzed the Chiari osteotomy [21], one the triple osteotomy [38], and one the combination of shelf acetabuloplasty + FVO [35].

### 3.3. Herring Classification Related to Stulberg Classification

Following Herring classification, 70% of B-type hips had an excellent surgical outcome, 22% had a fair outcome, and only 7% of them had a poor surgical outcome; the B/C group reported 57% excellent surgical outcome, 30% fair, and 14% poor; group C hips had an excellent surgical outcome in 38% of cases, fair in 35%, and poor in 26% (Figure 3).

### 3.4. Radiographic Stulberg Outcome of the Single Surgical Techniques

The extracted data reveals that an excellent surgical outcome, Stulberg I, occurs for FVO and Salter. In addition, Chiari osteotomy, now obsolete, and arthrodiatasis appear to have suitable surgical outcomes with a suitable Stulberg II percentage. The combined techniques (FVO + Shelf and FVO + Salter), on the other hand, seem to have more unfavorable outcomes with higher percentages of Stulberg III and IV (Figure 4).

### 3.5. Radiographic Stulberg Outcome Correct for Age at Onset

Correlating the surgical outcome to the age at onset, 6–8 years or more than 8 years, the surgical outcome for the two age groups was found to almost overlap, with a slight advantage for the 6–8 age group (Figure 5).

### 3.6. Arthroplasty Surgery

Only 7% of the hips needed prosthetic replacement under the age of 50 years, referring instead to the single surgical technique 7% for FVO, 1% for Shelf, 3% for FVO + Salter, and 4% for CO.

## 4. Discussion

This systematic review aims to clarify the surgical outcomes of Perthes patients in correlation with both the age of onset of the disease, the initial severity, and the surgical technique used.

The treatment of LCPD is still lacking official guidelines to date. However, there is a common trend in the treatment of LCPD that sees the use of conservative treatment in patients <6 years, resorting to surgical treatment in severe cases of disease and late-onset [41].

Nowadays, there is general agreement that surgery should aim to prevent loss of joint congruence by restoring the epiphysis to its central position within the acetabular cup. Many surgical techniques are used to date, and their indications differ [21].

FVO and Salter’s osteotomies are currently the most widely used surgical techniques; however, to obtain satisfactory results, it is necessary that the surgical intervention is carried out in the early phase of the fragmentation stage moreover and that the patient has a suitable range of motion of the hip before surgery, an objective that can also be achieved with the use of traction [15]; however, in the most severe forms, the combination of both or the use of other pelvic osteotomies are indicated [15].

The results support FVO in patients older than 6 years at onset, with more than 50% of the epiphysis affected. FVO is classically criticized as worsening the lower limb length deficiency, limiting hip abduction, and causing persistent limping [19]. Whether these criticisms are warranted remains strongly debated [14,17].

A review of the literature in the study by Braito et al. [42] concluded that femoral osteotomies are more frequently used than pelvic osteotomies. In the study by Herring et al. [43], both surgical techniques in patients of group B, B/C, aged more than 8 years, have excellent outcomes. Group C patients continue to have a poor prognosis. For Wiig et al., on the other hand, the ideal starting age for an excellent surgical outcome is 6 years or more [44].

According to the study by Maleki et al., patients over 8 years old might often have poor prognoses; however, advanced surgical and salvage surgical techniques provide suitable patient outcomes. In patients 6 to 8 years of age, the prognosis is variable and depends on the severity of the disease. Patients under 6 years of age have an excellent prognosis comparable to conservative treatment [45].

Salter’s innominate osteotomy was the first pelvic redirection osteotomy used in LCPD in 1962 [12]. Herring et al. [43] reported similar outcomes of LCPD at skeletal maturity with Salter’s osteotomy and FVO. In severe LCPD, however, Salter’s osteotomy does not provide sufficient femoral head coverage and may induce iatrogenic femoral-acetabular impingement [12]. The combination of Salter’s innominate osteotomy and femoral varus osteotomy is used when only one of these procedures is insufficient to cover the femoral head within the acetabulum. The principal indications for using these techniques are late clinical onset, severe femoral head involvement, evidence of lateral subluxation, lateral calcification, or considerable changes in the metaphysis [45,46,47].

The arthodiastasis and the transtrochanteric rotational osteotomy represent the most recently used femoral surgical treatments for LCPD. The first is considered as an alternative treatment, using an external fixator. It allows a mobile hip joint to keep the femoral head within the acetabulum without mechanical stress. The treatment aims to restore the articularity of the hip by avoiding contact between the femoral head and the acetabulum, promoting the regeneration and repair of damaged articular cartilage. It is used in patients with reduced range of motion and unmanageable pain; however, according to the analysis conducted by Leourox et al. [15], it should be abandoned as it is difficult to manage and with no superior results to other surgical techniques.

The transtrochanteric rotational osteotomy is a new surgical treatment used for children with late-onset LCPD after 9 years old [31,32].

Finally, Chiari osteotomy [21], a supracetabular rotational dislocation osteotomy in which the femoral head and joint capsule are mediated and covered by the inferior surface of the osteotomized ileal bone, has been almost completely abandoned [21]. This very cumbersome procedure is now reserved for the salvage treatment for more severe hips with lateral subluxation, severe femoroacetabular incongruity, or persistent painful hip. However, even in this situation, shelf acetabuloplasty and triple innominate osteotomy may be preferable [15]. Triple innominate osteotomy [38], which provides joint load reduction through hip medialization, and lateral shelf acetabuloplasty [28], a technique that allows increasing the superolateral coverage of the femoral head using a bone flap, is used as a pelvic rescue procedure in patients with age of onset >8 years and severe disease.

Lateral shelf acetabuloplasty became popular in the 1990s due to the simpler post-operative course and shorter time to weight-bearing compared to FVO. To date, there are still no large-scale studies on this subject in the literature. Suitable outcomes have been reported, however, even in patients with severe LCPD [27,36,37]

In all cases, the rate for an arthroplasty under 50 years old is 7%, referring instead to the single surgical technique 7% for FVO, 1% for Shelf, 3% for FVO + Salter, and 4% for CO.

In 1983, Perpich et al. [48] found a need for THA in 7% of their conservatively treated patients; in 1984, McAndrew et al. reported a need for THA in 40% [49]; in 2002, Lecuire et al. reported 24% [9]; and in 2011, Froberg et al. reported 13% [50].

To our knowledge, our study is the first to evaluate the surgical outcome of LCPD, taking into account all surgical techniques (single, combined, or save) rather than focusing only on the Salter and FVO osteotomy and setting a minimum limit of 5-year follow-up. Limitations of the study are an unspecified radiographic phase of initiation of treatment, heterogenicity of the data, and the study’s retrospective nature.

## 5. Conclusions

In conclusion, surgical treatment in patients aged more than 6 years was excellent in Herring B and B/C hips, with a high percentage of Stulberg I and II, poor results in Herring C hips, with a high percentage of Stulberg IV and V, with a slight advantage for patients aged between 6 and 8 years. Despite the lack of community guidelines, the early age at onset and, therefore, timely surgical treatment, in the initial phase of the fragmentation stage or before, which provides more time available for remodeling the femoral head before reaching skeletal maturity, the degree of initial disease severity and a suitable pre-operative range of motion remain the main indicators of surgical outcome to date.

## Figures and Tables

**Figure 1 children-09-01121-f001:**
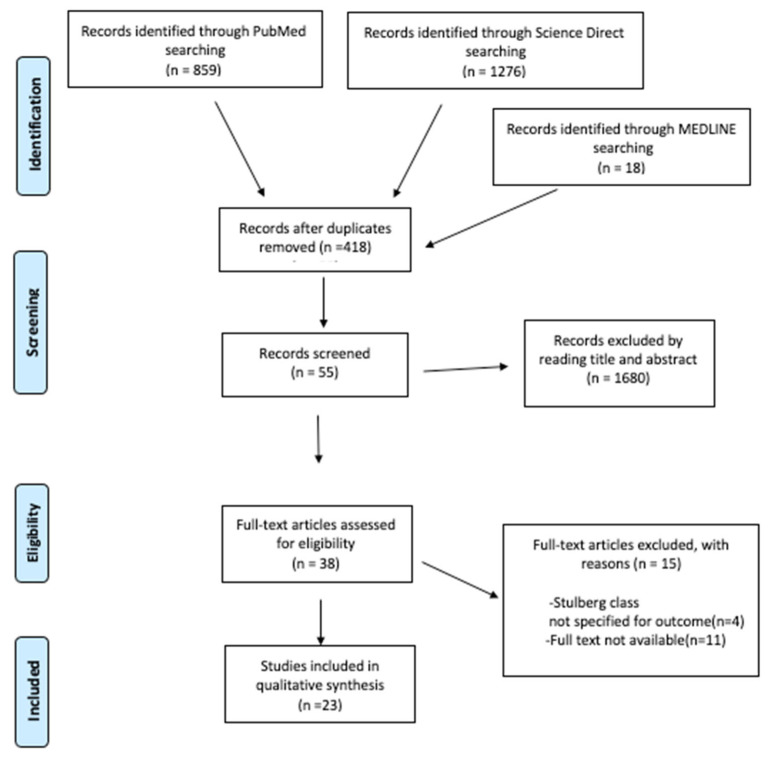
PRISMA (Preferred Reporting Items for Systematic Reviews and Meta-Analysis) flowchart of the systematic literature review.

**Figure 2 children-09-01121-f002:**
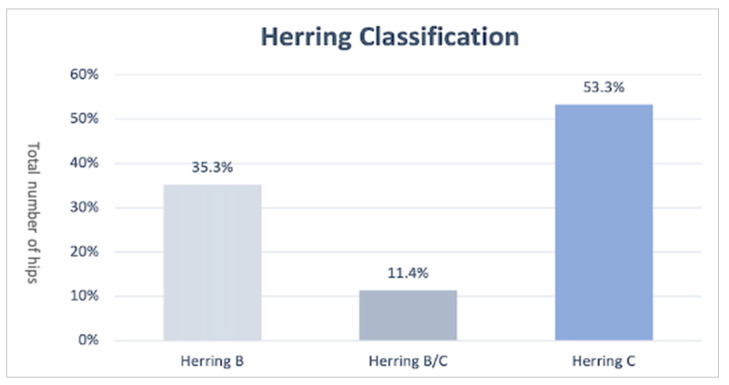
Herring classification.

**Figure 3 children-09-01121-f003:**
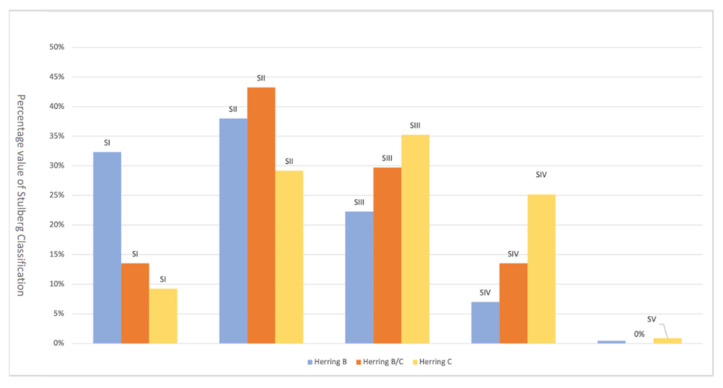
Herring classification related to Stulberg classification.

**Figure 4 children-09-01121-f004:**
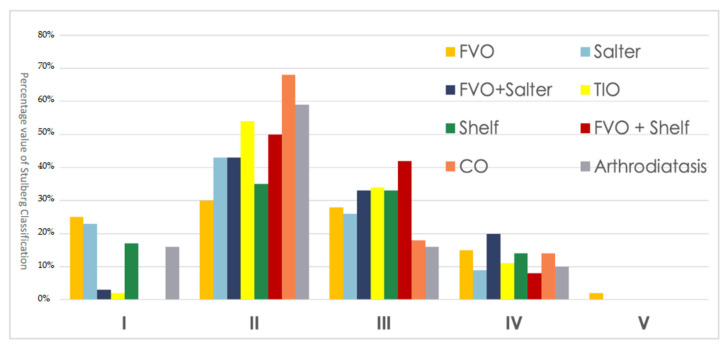
Radiographic Stulberg outcome of the single surgical techniques. FVO = femoral varus osteotomy; CO = Chiari osteotomy; TIO = triple innominate osteotomy; Shelf = lateral shelf acetabuloplasty.

**Figure 5 children-09-01121-f005:**
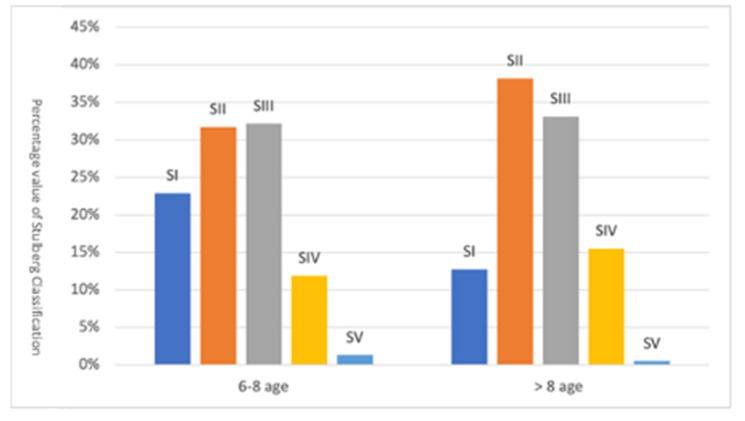
Radiographic Stulberg outcome correct for age at onset.

**Table 1 children-09-01121-t001:** Studies included in the systematic review.

Ref	Author	N. of Hip	Surgical Treatment	FU (y)	Stulberg Class					N. of THA
					I	II	III	IV	V	
[17]	Friedlander JK et al. (2000)	98	FVO	6.9	31	19	39	7	2	8
[18]	Talkhani IS et al. (2001)	16	FVO	7	3	8	3	2	0	2
[19]	Noonan KJ et al. (2001)	18	FVO	10	3	3	4	8	0	1
[20]	Van Der Geest IC et al. (2001)	21	Shelf	12	3	6	10	2	0	1
[21]	Reddy RR et al. (2005)	22	CO	6.1	0	15	4	3	0	1
[22]	Aksoy MC et al. (2005)	26	FVO	13.3	2	4	14	5	1	1
[23]	Sarassa CA et al. (2008)	10	FVO + SO	7	1	5	3	1	0	1
[24]	Freeman RT et al. (2008)	27	Shelf	5.2	14	0	10	3	0	1
[25]	Aly TA et al. (2009)	22	Arthrodiatasis	7	0	20	1	1	0	-
[26]	Javid M et al. (2009)	20	FVO + SO	5.5	0	6	9	5	0	-
[27]	Yoo WJ et al. (2009)	25	Shelf	6.7	1	8	13	3	0	-
[28]	Ghanem I et al. (2010)	30	Shelf	9.5	5	14	6	5	0	-
[29]	Pecquery R et al. (2010)	21	Shelf	4.3	2	12	1	5	1	-
[30]	Hosny GA et al. (2011)	29	Arthrodiatasis	7.5	8	9	7	4	1	-
[31]	Nakashima Y et al. (2011)	14	TRO	12	0	5	2	7	0	-
[32]	Farsetti P et al. (2012)	16	TRO	6.5	0	2	10	4	0	-
[33]	Grzegorzewski A et al. (2013)	23	Shelf	5.8	2	13	6	2	0	-
[34]	Yavuz U et al. (2013)	18	SO	6.5	2	8	7	1	0	-
[35]	Lim KS et al. (2015)	12	Shelf + FVO	10.1	0	4	7	1	0	-
[36]	Carsi B et al. (2015)	45	Shelf	11	4	12	21	7	0	-
[37]	Li WC et al. (2016)	51	Shelf	11	11	19	14	7	0	-
[38]	Stepanovich M et al. (2017)	56	TIO	8.5	1	35	14	6	0	-
[39]	Park KS et al. (2017)	29	SO	12.9	9	9	8	3	0	-

FU = Follow-up; CO = Chiari osteotomy; SO = Salter osteotomy; TRO = transtrochanteric rotational osteotomy; TIO = triple innominate osteotomy; FVO = femoral varus osteotomy; Shelf = lateral shelf acetabuloplasty.

## Data Availability

Not applicable.

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
