# Peer review of "Mid–Long-Term Outcomes of Surgical Treatment of Legg-Calvè-Perthes Disease: A Systematic Review"

_children, 2022, doi:10.3390/children9081121_

Round 1
Reviewer 1 Report
Authors need to enforce the paper -My comments are attached

Author Response
Thank you for the review. I added the citations you recommended and appreciated the comments made.
Please see the attachment

Reviewer 2 Report
I have reviewed article no. children-1810002 with the title ‘’Mid-long term outcomes of surgical treatment of Legg-Calvè-Perthes disease: a systematic review’’.
A few points need clarification see my main questions and specific comments.
main questions / main comments
1# First of all, I am interested in the lack of registration of the literature review. Why didn't the authors register the literature review e.g. in the PROSPERO database.
2# L80-81 – ‘’ The study was conducted in January 2022 using PubMed and Science Direct databases.” - Do I understand correctly the whole analysis was done in one month?
3# L80-81 – ‘’ The study was conducted in January 2022 using PubMed and Science Direct databases.” - Why were only two bases used ? Why exactly these? Didn't this significantly narrow the search for works?
4# - How the authors avoided the risk of bias? Please explain why the authors did not use the QUADAS-2 tool ?
5# - Why didn't the authors decide to do a statistical analysis of the data obtained, e.g. Meta-analysis ? I understand that authors would have to narrow down the searches even further, but it would have more evidentiary power.
6# - Discussion - there is a lack of explanations based a anatomy, physiology of the body of children. The authors focus too much on comparisons between the results of other researchers. Please add explanations of the results obtained based on anatomical, physiological and biochemical considerations.
specific comments
L4-5 - Why ‘’Alessia Caldaci1”, and ‘’and” are marked? Remove the check mark.
L44 – ‘’tion)[9]” - add spaces.
L53 – ‘’used to treat LCPD in 1962.” - add a citation to confirm the sentence.
L71 – L72 – ‘’A PIOS-approach [Patient (P); Intervention (I); Outcome (O); and Study Design (S)] 71 was utilized to carry out the study.’’ - add a citation to confirm the methods.
L82 – ‘’ The analysis of the texts was carried out by two independent operators.’’ - Please identify the authors doing the analysis.
L86 - Please add clear inclusion criteria to the paragraph.
L86 - The table title should be at the top. Graphic title at bottom.
Table 1 - Why is item number 16 in bold ?
L112 – ‘’ classification[ 10],’’ - Please correct your citation form and add spaces.
Figure 2., 3. and 5. - Please add a description of the percentage axis (Y axis) on the chart. What does it refer to?
Figure 4. - Add the x-axis and y-axis description on the graph.
Figure 4. - Underneath, expand the abbreviations used.
Figure 5. - Please add a description of the percentage axis (Y axis) on the chart. What does it refer to?
Figure 5. - Underneath, expand the abbreviations used. Each chart should be treated separately and the abbreviations used should be dispelled new.
L168 – ‘’ correlation ‘’ - The word is not appropriate. It indicates a form of statistical analysis. This word should be replaced.
L182 – ‘’ [14] [16].’’ - wrong form of quotation.
L225 – ‘’[27] [36] [50]’’ - wrong form of quotation.
L251 – 257 - Add the period at the end of each sentence.
Article 6 is a self-citation however, in my opinion, it is acceptable.
Thank you for the opportunity to review.
Author Response
1# First of all, I am interested in the lack of registration of the literature review. Why didn't the authors register the literature review e.g. in the PROSPERO database.
A1:It was not registered because the analysis started before the PROSPERO protocol was required, but we authors ensure that all the criteria and limits set by PROSPERO have been respected.
2# L80-81 – ‘’ The study was conducted in January 2022 using PubMed and Science Direct databases.” - Do I understand correctly the whole analysis was done in one month?
3# L80-81 – ‘’ The study was conducted in January 2022 using PubMed and Science Direct databases.” - Why were only two bases used ? Why exactly these? Didn't this significantly narrow the search for works?
A2+A3: Yes, we did it in a month and we preferred to use these two databases exclusively because they are the richest in terms of articles and above all because by adding other databases the number of duplicates would have increased and this would have lengthened the working time without effective gain.
4# - How the authors avoided the risk of bias? Please explain why the authors did not use the QUADAS-2 tool ?
A4:The QUADAS2 is an excellent tool, however even the one we use is commonly accepted and we chose it because we were more familiar with it in order to avoid selection bias errors.
5# - Why didn't the authors decide to do a statistical analysis of the data obtained, e.g. Meta-analysis ? I understand that authors would have to narrow down the searches even further, but it would have more evidentiary power.
A5: Due to the nature of the articles that are not randomized clinical trials, but of a retrospective nature, the meta-analysis could have given misleading results, therefore we opted for a systematic review.
6# - Discussion - there is a lack of explanations based a anatomy, physiology of the body of children. The authors focus too much on comparisons between the results of other researchers. Please add explanations of the results obtained based on anatomical, physiological and biochemical considerations.
A6: Excellent observation, but it was not possible because not all the articles examined contain the required data
For specific comments, please see the attachment

Round 2
Reviewer 1 Report
The changes are convincing.
Author Response
Thanks to the availability
Reviewer 2 Report
Thank you for sending the revised version for re-evaluation. I appreciate the authors' work in improving the manuscript.
main questions / main comments
A4-A5 - The authors' explanation is sufficient and acceptable.
A1-A3 and A6 -I regret to say that I do not accept the authors' explanation.
"A1:It was not registered because the analysis started before the PROSPERO protocol was required, but we authors ensure that all the criteria and limits set by PROSPERO have been respected."
I understand but this is an error related to the planning of the systematic review.
" A2+A3: Yes, we did it in a month and we preferred to use these two databases exclusively because they are the richest in terms of articles and above all because by adding other databases the number of duplicates would have increased and this would have lengthened the working time without effective gain."
I agree that adding bases would increase the time. In my opinion, it is worthwhile in science sometimes to carry out the work more slowly and accurately. I would disagree with the statement that it would be inefficient. Removing duplicates is a natural result of conducting a review of several bases.
"A6: Excellent observation, but it was not possible because not all the articles examined contain the required data."
I can't agree with that. The authors can make hypothetical considerations based on anatomical and physiology knowledge.
specific comments
Most of the specific comments have been well-corrected.
I am not satisfied with the improvement of the charts. In my opinion, the description of the axis should be on the graphic and not under it.
Other minor comments that occurred to me after re-examining the manuscript:
L46 - change "[8,9,10,11,12,13,14,15,16,17,18] " to "[8-18]"
Figure 4. - Add the unit (%) on the Y axis.
Author Response
main questions / main comments
A1-A3 and A6 -I regret to say that I do not accept the authors' explanation.
"A1:It was not registered because the analysis started before the PROSPERO protocol was required, but we authors ensure that all the criteria and limits set by PROSPERO have been respected."
I understand but this is an error related to the planning of the systematic review.
R:In agreement with the editor who agrees that it is not useful to register with PROSPERO for this work, I have added the description of the PRISMA statement in the materials and methods section as per his request
" A2+A3: Yes, we did it in a month and we preferred to use these two databases exclusively because they are the richest in terms of articles and above all because by adding other databases the number of duplicates would have increased and this would have lengthened the working time without effective gain."
I agree that adding bases would increase the time. In my opinion, it is worthwhile in science sometimes to carry out the work more slowly and accurately. I would disagree with the statement that it would be inefficient. Removing duplicates is a natural result of conducting a review of several bases.
R: According to your request, I expanded the search with MEDLINE without any new results (see materials and methods)
"A6: Excellent observation, but it was not possible because not all the articles examined contain the required data."
I can't agree with that. The authors can make hypothetical considerations based on anatomical and physiology knowledge.
R:I modified the discussion and conclusions section by adapting the outcomes by surgical technique, objectives of our study, to the biomechanical, clinical and radiographic characteristics of the patientsall specific comments were satisfied (see attached file)
All specific comments were satisfied (see attached file)
